# Differences in the Protection Motivation Theory Constructs between People with Various Latent Classes of Motivation for Vaccination and Preventive Behaviors against COVID-19 in Taiwan

**DOI:** 10.3390/ijerph18137042

**Published:** 2021-07-01

**Authors:** Yi-Lung Chen, Yen-Ju Lin, Yu-Ping Chang, Wen-Jiun Chou, Cheng-Fang Yen

**Affiliations:** 1Department of Healthcare Administration, Asia University, Taichung 41354, Taiwan; elong@asia.edu.tw; 2Department of Psychology, Asia University, Taichung 41354, Taiwan; 3Department of Psychiatry, School of Medicine, College of Medicine, Kaohsiung Medical University, Kaohsiung 80708, Taiwan; 1040457@kmuh.org.tw; 4Department of Psychiatry, Kaohsiung Medical University Hospital, Kaohsiung 80708, Taiwan; 5School of Nursing, The State University of New York at Buffalo, New York, NY 14214-3079, USA; yc73@buffalo.edu; 6School of Medicine, Chang Gung University, Taoyuan 33302, Taiwan; 7Department of Child and Adolescent Psychiatry, Chang Gung Memorial Hospital, Kaohsiung Medical Center, Kaohsiung 83301, Taiwan

**Keywords:** COVID-19, vaccine, motivation, preventive behavior, Protection Motivation Theory

## Abstract

The present study aimed to identify the distinct classes of motivations to get vaccinated and to adopt preventive behaviors against the coronavirus disease 2019 (COVID-19) among people in Taiwan and to examine the roles of Protection Motivation Theory (PMT) cognitive constructs in these unique classes of motivations to get vaccinated and to adopt preventive behaviors. We recruited 1047 participants by using a Facebook advertisement. Participants’ motivations to get vaccinated and to adopt preventive behaviors against COVID-19, PMT constructs of threat appraisal (perceived severity and perceived vulnerability), and PMT constructs of coping appraisal (self-efficacy, response efficacy, response cost, knowledge, and previous vaccination for seasonal influenza) were determined. We analyzed participants’ motivations to get vaccinated and to adopt preventive behaviors against COVID-19 by using latent profile analysis. The present study identified three latent classes, including the participants with high motivation for vaccination and preventive behaviors (the class of Both High), those with low motivation for vaccination and preventive behaviors (the class of Both Low), and those with high motivation for vaccination but low motivation for preventive behaviors (the class of High Vaccination but Low Preventive Behaviors). Compared with the participants in the class of Both High, participants in the class of Both Low had lower levels of perceived vulnerability, perceived severity, self-efficacy to have vaccination, response efficacy of vaccination, knowledge about vaccination, and previous vaccination for seasonal influenza; participants in the class of High Vaccination but Low Preventive Behaviors had lower levels of perceived vulnerability and perceived severity but higher levels of response cost of vaccination. We concluded that varieties of motivations, threat, and coping appraisals should be considered in intervention programs aiming to increase motivation to adopt recommended protective behaviors against COVID-19.

## 1. Introduction

### 1.1. Vaccination and Preventive Behaviors against Coronavirus Disease 2019

Coronavirus disease 2019 (COVID-19) has had a disastrous effect worldwide [1,2,3,4,5,6,7,8]. Governments must take urgent actions to curb the spread of the highly contagious COVID-19. An efficacious vaccine is considered essential to prevent further morbidity and mortality of COVID-19 infection [9]. COVID-19 vaccines have been developed quickly, and they are expected to stop the spread of COVID-19 [10,11]. Seven vaccines are currently approved by the World Health Organization for emergent use against COVID-19 [12]. Although data support the safety and efficacy of these vaccines [13,14,15,16,17], uncertainty regarding the effects of vaccines on new COVID-19 variants [18] and provision of these vaccines to low-income countries [19,20] are imminent challenges. Until COVID-19 vaccines are made available to all, several preventive behaviors, such as social distancing, wearing face masks, and washing hands regularly, are recommended to reduce the transmission of COVID-19 [10,21,22,23]. Therefore, preventive behaviors remain vital for reducing the risk of contracting COVID-19.

### 1.2. Motivation to Get Vaccinated and to Adopt Preventive Behaviors against COVID-19

Although adopting preventive behaviors and getting vaccinated against COVID-19 are crucial during the COVID-19 pandemic, motivation to implement them is not universal. Many individuals have resisted adherence to social distancing [24], wearing a mask [25], and washing their hands regularly [26]. Moreover, an unfavorable attitude toward getting vaccinated against COVID-19 was reported in 16.7%, 14.8%, and 9% of the population in China [27], the USA [28], and the UK [29], respectively. For developing intervention programs to reduce the COVID-19 risk, factors that motivate people to get vaccinated and to adopt preventive behaviors must be predicted.

Research has demonstrated a significant association between the adoption of preventive behaviors against COVID-19 and motivation to receive a COVID-19 vaccination. However, adopting preventive behaviors did not guarantee a high motivation to receive a COVID-19 vaccination [30,31]. Most of the recommended preventive behaviors against COVID-19 are well-known measures to prevent the spread of respiratory infectious diseases with proven benefits; however, the COVID-19 vaccines have been developed in a short period, causing some individuals to be skeptical. Conversely, preventive behaviors may cause inconveniences in daily life; therefore, individuals may expect vaccination to fundamentally reduce the threat of COVID-19. Identifying the distinct classes of motivation to get vaccinated and to adopt preventive behaviors and examining the factors related to various classes of motivations may provide knowledge to develop prevention strategies against COVID-19.

### 1.3. Applying Protection Motivation Theory to Evaluate the Motivation to Get Vaccinated and to Adopt Preventive Behaviors

Protection Motivation Theory (PMT) is one of the theoretical frameworks that has been commonly applied to explain the motivation and behaviors regarding vaccination for respiratory infectious diseases [32,33,34,35,36,37]. PMT consists of two cognitive processes: threat appraisal and coping appraisal [34]; both have essential roles in determining the level of motivation to adopt certain behaviors to protect individuals from contracting respiratory infectious diseases [35]. Threat appraisal is composed of the perceived severity of and vulnerability to respiratory infectious diseases; coping appraisal is composed of perceived self-efficacy, response efficacy, and cost efficacy of adopting protective behaviors [35,36,38]. The result of a review study supported the effectiveness of Protection Motivation Theory, Health Belief Model, and Theory of Planned Behavior on informing infectious disease modelling, research, and public health practice [39]. Because knowledge regarding COVID-19 and protective behaviors [40,41] and previous experiences regarding adopting protective behaviors [31] also predict further adoption of protective behaviors against respiratory infectious diseases, they are included in the extended PMT.

PMT has been applied to explain the motivation to adopt preventive behaviors [41,42,43,44,45,46] and to get vaccinated against COVID-19 [31,44,47,48]. Cross-sectional studies have revealed that perceived severity of and vulnerability to COVID-19, self-efficacy to adopt preventive behaviors, and response efficacy of preventive behaviors are positively associated with the intention to adhere to recommended protective behaviors against COVID-19 [41,42,43,44,45]. A follow-up study demonstrated that perceived severity and response efficacy can predict the adoption of preventive behaviors against COVID-19 [44]. Moreover, research has revealed that the perceived severity of COVID-19 [46], response and cost efficacy of vaccination [44,48], and receiving an influenza vaccine [31] are associated with the motivation to receive COVID-19 vaccination. However, no study has examined whether the cognitive constructs of PMT differ between individuals with various classes of motivations to get vaccinated and to adopt preventive behaviors against COVID-19. If the PMT constructs are different, the programs for controlling the COVID-19 pandemic should be tailored according to these classes.

### 1.4. Study Aims and Hypotheses

The present study aimed (1) to identify the distinct classes of motivations to get vaccinated and to adopt preventive behaviors against COVID-19 among people in Taiwan and (2) to examine the roles of PMT cognitive constructs in these unique classes of motivations to get vaccinated and to adopt preventive behaviors. We proposed two hypotheses. First, because adopting preventive behaviors did not guarantee high motivation to receive COVID-19 vaccination [30,31], we hypothesized that the individuals could be categorized into various latent classes according to their motivations of getting vaccinated and adopting preventive behaviors against COVID-19. Second, we hypothesized that the cognitive constructs of PMT differ between individuals with various classes of motivations to get vaccinated and to adopt preventive behaviors against COVID-19.

## 2. Methods

### 2.1. Participants

The procedure of recruiting participants in this study is described elsewhere [49]. In brief, 1047 participants were recruited through a Facebook advertisement between 15 October 2020 and 21 December 2020. The inclusion criteria were age ≥20 years and living in Taiwan. As of 21 December 2020, in Taiwan, 627 patients had COVID-19 and 7 patients had died [50]. No vaccine against COVID-19 was available in Taiwan during the study period. The Institutional Review Board of Kaohsiung Medical University Hospital approved this study (KMUHIRB-EXEMPT (I) 20200019).

### 2.2. Measures

#### 2.2.1. Motivation to Get Vaccinated for COVID-19

The motivation to get vaccinated for COVID-19 was assessed using one item. The question and scoring are listed in Table 1. A higher score indicated higher motivation to get vaccinated for COVID-19 [47].

#### 2.2.2. Motivation to Adopt Preventive Behaviors

Three items were adopted from the questionnaire developed by Liao et al. [51] to measure the adoption of preventive behaviors (avoiding crowded places, washing hands regularly, and wearing a mask) to protect against COVID-19 in the past week. The questions and scores are listed in Table 1. A higher total score indicated higher motivation to adopt preventive behaviors.

#### 2.2.3. Constructs of the Extended PMT

We measured the constructs of threat appraisal (perceived severity of COVID-19 and perceived vulnerability to COVID-19) by the questionnaire transformed from the questionnaire for measuring risk perception of the H1N1 influenza [51]. A previous study using this questionnaire found that individuals who adopted protective behaviors against COVID-19 had higher levels of threat appraisal than who did not (*p* < 0.001) [52] and supported its discriminant validity. We measured the constructs of coping appraisal (self-efficacy of receiving COVID-19 vaccination, response efficacy of COVID-19 vaccination, response cost of COVID-19 vaccination, and knowledge about COVID-19 vaccination) by the items of the Drivers of COVID-19 Vaccination Acceptance Scale (DrVac-COVID19S) [53]. A previous study found that the DrVac-COVID19S had acceptable validity and could quickly capture the individuals’ intrinsic intention receive COVID-19 vaccination [53]. The Cronbach α of threat appraisal and coping appraisal of this study was 0.704 and 0.821, respectively. Furthermore, the extended PMT included previous experience of vaccination for seasonal influenza. The items and scores are listed in Table 1. A higher score on each construct indicated higher protection motivation.

#### 2.2.4. Sociodemographic Characteristics

Sex (female vs. male), age, and education level (0 = *primary school or below*; 1 = *junior high school*; 2 = *senior high school*; 3 *= college or university*; 4 = *master’s degree*; and 5 = *doctorate*) were collected. Participants were divided into three groups according to age (<35, 35–49, and ≥50 years).

### 2.3. Statistical Analysis

We analyzed participants’ motivations to get vaccinated for COVID-19 and to adopt preventive behaviors against COVID-19 through Latent Profile Analysis (LPA) by using the R package *tidyLPA* [54] with the standardization of motivations to get vaccinated and to adopt preventive behaviors. The latent class obtained from the LPA was used to determine the motivations to get vaccinated for COVID-19 and to adopt preventive behaviors against COVID-19 for latent class membership. The number of classes was selected based on the basic model according to four model fit indices: Akaike information criterion (AIC), Bayesian information criterion (BIC), entropy, and the bootstrapped likelihood ratio test (BLRT). Models with lower AIC and BIC have a better fit than those with higher AIC and BIC values. A value of entropy approaching 1 indicates a clear separation of classes [55], and entropy >0.80 indicates that the latent classes are highly discriminating [56]. For BLRT, *p* < 0.05 indicates that the *k* class model is superior to the *k* − 1 class model (*k* represents the number of classes).

The differences in sociodemographic characteristics between various latent classes of motivations to get vaccinated and to adopt preventive behaviors against COVID-19 were examined using multiple multinomial logistic regression with the latent class as the nominal outcome variable. The differences in the PMT constructs of perceived severity, perceived vulnerability, self-efficacy to vaccination, response efficacy and cost of vaccination, knowledge, and previous vaccination for seasonal influenza between the latent classes were examined using a multiple multinomial logistic regression with adjustment for sociodemographic characteristics.

## 3. Results

### 3.1. Results of LPA

Table 2 presents the result of model fit indices for the LPA analysis. The three-latent-class model (AIC = 5590.48.1, BIC = 5640.02, and BLRT = 0.01) was selected based on its minimal AIC and BIC values and BLRT < 0.05.

Figure 1 presents the standard scores of the motivation to receive a vaccination and adopt preventive behaviors against COVID-19 per latent class. The first latent class (39.8% of the sample, 417/1047) was named “high motivation for vaccination and preventive behaviors” and comprised participants with high scores in motivations for both vaccination and preventive behaviors. The second latent class (11.7% of the sample, 123/1984) was named “low motivation for vaccination and preventive behaviors” and consisted of those with low scores in both motivation for vaccination and preventive behaviors. The third latent class (48.4% of the sample, 507/1047) was named “high motivation for vaccination but low for preventive behaviors” and consisted of those with high scores for vaccination but low scores for protective behaviors.

### 3.2. PMT Constructs Predicting the Latent Classes

Table 3 presents the results of the multiple multinomial logistic regression analysis examining the differences in sociodemographic characteristics and PMT constructs between the latent classes of motivations to get vaccinated and to adopt protective behaviors. Compared with participants with high motivations for vaccination and preventive behaviors, participants with high motivations for vaccination but low for preventive behaviors were less likely to belong to the age groups of 35–49 or ≥50 years. No differences were observed in sex or education between the three latent classes.

After adjustment for sociodemographic characteristics, compared with participants with high motivations for vaccination and preventive behaviors, participants with low motivation for vaccination and preventive behaviors had lower levels of perceived vulnerability, perceived severity, self-efficacy to have vaccination, response efficacy of vaccination, knowledge about the vaccination, and previous vaccination for seasonal influenza. Compared with participants with high motivations for vaccination and preventive behaviors, participants with high motivations for vaccination but low for preventive behaviors had lower levels of perceived vulnerability and perceived severity but a higher level of response cost of vaccination.

## 4. Discussion

The present study identified three latent classes according to the motivation to get vaccinated and to adopt preventive behaviors against COVID-19, namely high motivation for vaccination and preventive behaviors, low motivation for vaccination and preventive behaviors, and high motivation for vaccination but low for preventive behaviors. Several PMT cognitive constructs differed between these latent classes. Research has supported the prediction of motivation to get vaccinated for further actual behaviors of vaccination. For example, a one-year follow-up study confirmed that parents’ lower intention to have their children vaccinated significantly predicted the lower likelihood of influenza vaccination among children [57]. A nine-month follow-up study confirmed that higher vaccination intention predicted greater vaccination uptake in healthcare workers [58]. Therefore, enhancing people’s motivation to get vaccinated and to adopt preventive behaviors against COVID-19 is important to stop the spread of COVID-19.

### 4.1. Low Motivation for Getting Vaccinated and Adopting Preventive Behaviors and Related PMT Constructs

The present study classified participants into three latent classes based on their motivations to get vaccinated and to adopt preventive behaviors against COVID-19. Participants with low motivation for getting vaccinated and adopting preventive behaviors accounted for 11.7% of the population; this group perceived lower vulnerability to COVID-19 and severity of COVID-19. Moreover, they reported lower self-efficacy to, response efficacy of, and knowledge regarding COVID-19 vaccination, and fewer previous experiences of vaccination for seasonal influenza compared with those with high motivations for getting vaccinated and adopting preventive behaviors. Lack of basic protection may put this group at the highest risk of contracting COVID-19. This group of the individuals should be the target of intervention programs for controlling the spread of COVID-19. Enhancing their threat appraisal toward COVID-19 and coping appraisal toward vaccination should be important components of intervention programs. However, despite media reports regarding the highly contagiousness and lethal nature of COVID-19 since February 2020, this group retained low motivation. Their cognitive processing model of risk perception warrants further study to provide evidence for developing effective intervention programs. Taiwan has succeeded in mass-vaccination programs against seasonal influenza [59] and hepatitis B [60]. Lessons learned from the previous mass-vaccination planning may provide recommendations for enhancing preparedness for the COVID-19 emergency that requires mass vaccination.

### 4.2. High Motivation for Getting Vaccination but Low Motivation for Adopting Preventive Behaviors and Related PMT Constructs

The present study demonstrated that nearly half (48.4%) of the participants had high motivation to get vaccinated but low motivation to adopt preventive behaviors against COVID-19. They had lower levels of perceived vulnerability to and severity of COVID-19 but a higher level of response cost of vaccination compared with those with high motivations to get vaccinated and to adopt preventive behaviors. Low threat appraisal might partially account for their low motivation to adopt preventive behaviors against COVID-19. Moreover, low motivation to adopt preventive behaviors might result from repeated alerts regarding the COVID-19 pandemic but the low number of COVID-19 cases in Taiwan in the past 6 months. People might consider that getting vaccinated for COVID-19 is an alternative to adopting preventive behaviors in daily lives. Moreover, annual vaccination for seasonal influenza among people in Taiwan [59] may contribute to their high motivation to get vaccinated for COVID-19. A high level of response cost of vaccination indicates a high level of concern regarding money, time, and effort associated with receiving a vaccination for COVID-19.

The high proportion of people with high motivation to get vaccinated but low motivation to adopt protective behaviors against COVID-19 is a warning regarding the poor prevention strategy in Taiwan. Although vaccination is the fundamental strategy to restrain the spread of COVID-19 [10,11], no COVID-19 vaccine can completely protect people from being infected. Many countries including Taiwan are encountering the difficulty of insufficient supply of COVID-19 vaccines [61]. The government in Taiwan announced the latest COVID-19 vaccine distribution strategy on 21 June 2021 by distributing the limited COVID-19 vaccines to the 10 priority groups, with the first three groups to receive doses immediately; the 10 approved priority groups are: healthcare workers (Group 1), central and local government epidemic prevention personnel (Group 2), frontline workers with a high risk of coming into contact with COVID-19 (Group 3), those who need to travel abroad (Group 4), law enforcement officers and firefighters (Group 5), volunteers, long-term caretakers, and care recipients at social welfare organizations (Group 6), national security personnel (Group 7), adults aged 65 and above (Group 8), adults aged 19 to 65 with life-threatening conditions, rare diseases, or a history of serious illness (Group 9), and adults between the ages of 50 and 64 (Group 10) [62]. This indicates that many people may not obtain the protection of vaccination against COVID-19 until their priority group is permitted to receive a vaccination. It is important for all people to continue implementing the recommended protective behaviors to reduce the risk of contracting COVID-19.

The present study found that, compared with participants with high motivations for vaccination and preventive behaviors, participants with high motivations for vaccination but low for preventive behaviors were more likely to belong to the age group of <35, whereas no difference in age distribution between the classes of both high and both low motivations for vaccination and preventive behaviors was found. Older age is identified as the major risk factor for COVID-19 [63]. A study examining elderly people’s responses to the COVID-19 pandemic using data from 27 countries revealed that elderly people’s compliance with preventive measures is substantially similar to their fellow citizens in their 50′s and 60′s [64]. The results of the present and previous studies indicated that governments must improve their strategies to remind elderly people of their vulnerability to COVID-19 and the necessity of complying with preventive measures.

### 4.3. Strengths and Limitations

The present study is one of the first studies to classify individuals with various motivation levels to get vaccinated and to adopt protective behaviors against COVID-19 and to examine differences in PMT cognitive constructs across the individuals of various classes of motivation. The results can provide knowledge to develop intervention programs to subtly enhance motivation of people in various groups to adopt all types of protective behaviors. However, the present study had some limitations. First, this study used a Facebook advertisement to recruit participants. Although delivering study questionnaires through social media can provide a large number of responses in a short period [65], particularly during the pandemic, the problem of sample bias is inevitable [66]. For example, Facebook users consist of younger and more progressive people among the general population [56]. Moreover, social desirability bias may increase the possibility for the respondents to underreport their noncompliance with public health measures in the context of the COVID-19 pandemic [67]. Researchers found that the guilt-free strategy is a useful tool in increasing respondents’ proclivity to truthfully report their non-compliance with COVID-19 preventive measures [68]. We used online anonymous questionnaires to reduce the possibility of social desirability bias. Further studies are needed to examine whether online anonymous questionnaires can reduce the social desirability bias compared with registered questionnaires. Second, the PMT constructs of coping appraisal examined in this study focused on the participants’ attitudes toward getting vaccinated for COVID-19 but not toward adopting preventive behaviors. Third, we chose PMT as the theoretical framework to understand the participants’ appraisal and coping appraisal of vaccination against COVID-19; however, we did not evaluate the impacts of environmental variables such as social norms and government policies. Moreover, follow-up studies are needed to examine the psychological, socioeconomic, and political moderators and mediators of the association between motivation to and actual behaviors of vaccination against COVID-19. For example, research demonstrated that socio-economic factors influenced the public’s willingness to pay for the vaccine [27]. Fourth, we assessed participants’ knowledge about COVID-19 vaccination using the items on the DrVac-COVID19S; however, participants’ accuracy of knowledge about the mechanism of COVID-19 vaccination could not be confirmed. Further study should examine the individuals’ knowledge about the effectiveness and complications of various vaccines against COVID-19 and their associations with the motivation to get vaccinated.

## 5. Conclusions

The present study identified three latent classes of individuals with various motivations to get vaccinated and to adopt preventive behaviors against COVID-19. Furthermore, we determined the differences in threat and coping appraisals in terms of PMT between the latent classes. On the basis of the results, we suggest that designers of intervention programs aiming to increase motivation to adopt recommended protective behaviors against COVID-19 should consider the varieties of motivations, threat, and coping appraisals. Given that lack of vaccination and protective behaviors may put the people with low motivation for getting vaccinated and adopting preventive behaviors at the highest risk of contracting COVID-19, intervention programs should focus on enhancing their threat appraisal toward COVID-19 and coping appraisal toward vaccination. For people with high motivation to get vaccinated but low motivation to adopt preventive behaviors against COVID-19, enhancing their threat appraisal toward COVID-19 and discussing the advantages and disadvantages of vaccination against COVID-19 are necessary to ensure their vaccination against COVID-19.

## Figures and Tables

**Figure 1 ijerph-18-07042-f001:**
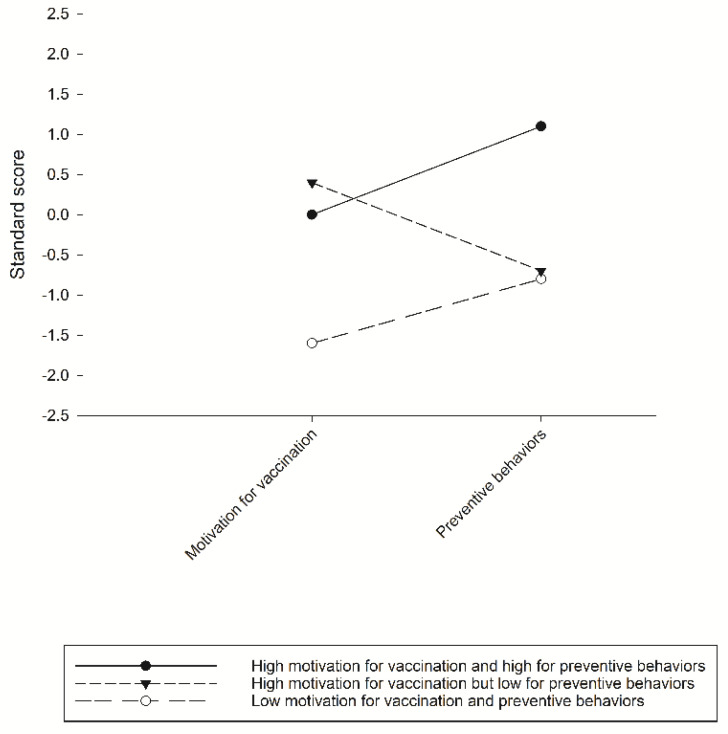
Three classes of participants with various levels of motivation to receive COVID-19 vaccination and to adopt preventive behaviors.

**Table 1 ijerph-18-07042-t001:** Motivation to Receive a COVID-19 Vaccination with Related Constructs of the Extended PMT.

Measures	Items	Response Scale
*Motivation to receive a COVID-19 vaccination*	Please rate your current willingness to receive a COVID-19 vaccine:	1 (*very low*) to 10 (*very high*)
*Motivation to adopt protective behaviors*	In the past week, did you (1) avoid going to crowded places, (2) wash your hands more often, and (3) wear a mask more often?	Each item was rated as 0 (*no*) or 1 (*yes*) and summed to obtain a total score
*Constructs of PMT*		
Perceived severity	Item 1: Please rate the current level of your concern about COVID-19:	1 (*very mild*) to 10 (*very severe*)
	Item 2: How serious is COVID-19 relative to SARS?	1 (*much less serious*) to 5 (*much more serious*)
Perceived vulnerability	Item 1: How likely do you think you are to contract COVID-19 over the next month?	1 (*very unlikely*) to 7 (*very likely*)
	Item 2: If you were to develop flu-like symptoms tomorrow, would you be worried?	1 (*not at all worried*) to 7 (*extremely worried*)
	Item 3: In the past week, how often have you worried about catching COVID-19?	1 (*never*) to 5 (*all the time*)
	Item 4: What do you think your chances are of getting COVID-19 over the next month are compared with others outside your family?	1 (*no chance*) to 7 (*certain*)
Self-efficacy of COVID-19 vaccination	I can choose whether to receive a COVID-19 jab or not.	1 (*strongly disagree*) to 7 (*strongly agree*)
Response efficacy of COVID-19 vaccination	Item 1: Vaccination is a very effective way to protect me against COVID-19.	1 (*strongly disagree*) to 7 (*strongly agree*)
	Item 2: It is important that I receive the COVID-19 vaccine.	
	Item 3: Vaccination greatly reduces my risk of catching COVID-19.	
	Item 4: The COVID-19 vaccine plays an important role in protecting my life and that of others.	
	Item 5: The COVID-19 shot will make an important contribution to my health and well-being.	
	Item 6: Receiving the COVID-19 shot has a positive influence on my health.	
Response cost of COVID-19 vaccination	(1) Safety and possible side effects of vaccine, (2) cost of vaccine, and (3) time spent on vaccination will influence my willingness to get vaccinated for COVID-19.	1 (*strongly disagree*) to 4 (*strongly agree*)
Knowledge about COVID-19 vaccination	Item 1: I understand how the COVID-19 shot helps my body fight the COVID-19 virus.	1 (*strongly disagree*) to 7 (*strongly agree*)
	Item 2: I understand how vaccination protects me from COVID-19.	
	Item 3: How the COVID-19 jab works to protect my health is a mystery to me. *	
Previous vaccination for seasonal influenza	Did you receive an influenza vaccination in recent years?	1 (*never*) to 4 (*always*)

PMT = Protection Motivation Theory; COVID-19 = coronavirus disease 2019. *: Reverse scoring.

**Table 2 ijerph-18-07042-t002:** Summary of Information for Selecting the Number of Latent Classes for Latent Profile Analysis.

No. of Classes	AIC	BIC	Entropy	BLRT (*p*-Value)
1	5948.51	5968.33	1	-
2	5954.41	5989.09	0.21	0.48
3	5590.48	5640.02	0.74	0.01
4	5596.53	5660.93	0.6	0.67
5	5497.76	5577.02	0.78	0.01
6	5497.01	5591.13	0.73	0.02

AIC = Akaike information criterion, BIC = Bayesian information criterion, BLRT = bootstrapped likelihood ratio test.

**Table 3 ijerph-18-07042-t003:** Comparisons of sociodemographic characteristics and constructs of PMT between latent classes of motivation to get vaccinated and to adopt preventive behaviors: multinomial logistic regression.

Variable	High Motivations for Vaccination and High for Preventive Behaviors (*N* = 417)	Low Motivations for Vaccination and Low for Preventive Behaviors (*N* = 123)	OR 1 ^c^ (95% CI)	High Motivations for Vaccination but Low for Preventive Behaviors (*N* = 507)	OR 2 ^c^ (95% CI)
*Sociodemographics*					
Gender ^a^					
Female	250 (60.0%)	81 (65.9%)	1.00	286 (56.4%)	1.00
Male	167 (40.0%)	42 (34.1%)	0.78 (0.51–1.18)	221 (43.6%)	1.16 (0.89–1.50)
Age ^a^					
<35	191 (45.8%)	62 (50.4%)	1.00	289 (57.0%)	1.00
35–49	175 (42.0%)	46 (37.4%)	0.81 (0.53–1.25)	187 (36.9%)	0.71 (0.54–0.93) *
≥50	51 (12.2%)	15 (12.2%)	0.91 (0.48–1. 72)	31 (6.1%)	0.40 (0.25–0.65) ***
Education levels ^a^					
High school or below	43 (10.3%)	13 (10.6%)	1.00	54 (10.7%)	1.00
Bachelor’s degree	247 (59.2%)	85 (69.1%)	1.14 (0.58–2.22)	335 (66.1%)	1.08 (0.70–1.67)
Master’s degree and above	127 (30.5%)	25 (20.3%)	0.65 (0.31–1.38)	118 (23.3%)	0.74 (0.46–1.19)
*PMT constructs* ^b^					
Perceived vulnerability	7.8 ± 3.3	5.6 ± 3.3	0.81 (0.76–0.87) ***	6.4 ± 3.1	0.88 (0.84–0.92) **
Perceived severity	7.6 ± 3.0	5.6 ± 2.9	0.79 (0.73–0.85) ***	6.3 ± 2.9	0.85 (0.81–0.89) **
Self-efficacy to have vaccination	5.0 ± 1.1	4.6 ± 1.8	0.75 (0.64–0.88) ***	5.0 ± 1.0	0.96 (0.85–1.09)
Response efficacy of vaccination	25.4 ± 6.8	15.7 ± 6.5	0.81 (0.78–0.84) ***	25.7 ± 5.6	1.01 (0.98–1.03)
Response cost of vaccination	5.6 ± 1.9	5.6 ± 2.0	0.96 (0.87–1.07)	6.0 ± 1.8	1.11 (1.04–1.19) **
Knowledge about vaccination	11.3 ± 4.0	8.8 ± 4.0	0.85 (0.80–0.90) **	11.9 ± 3.7	1.03 (1.00–1.07)
Previous vaccination for seasonal influenza	1.5 ± 1.3	1.0 ± 1.3	0.72 (0.61–0.85) ***	1.5 ± 1.3	1.02 (0.92–1.13)

^a^ Unadjusted multinomial logistic regression. ^b^ Multinomial logistic regression with the adjustment of sociodemographic characteristics. ^c^ The group with a high motivation to get vaccinated and to adopt preventive behaviors serves as the reference. CI: confidence interval; PMT = Protection Motivation Theory. * *p* < 0.05; ** *p* < 0.01; *** *p* < 0.001.

## Data Availability

The data are available on reasonable request to the corresponding authors.

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
