# Peer review of "Differences in the Protection Motivation Theory Constructs between People with Various Latent Classes of Motivation for Vaccination and Preventive Behaviors against COVID-19 in Taiwan"

_ijerph, 2021, doi:10.3390/ijerph18137042_

Round 1
Reviewer 1 Report
The manuscript of Chen et al. explores an important subject during this phase of the COVID-19 pandemic what motivates people to get vaccinated and what drives their behavior concerning preventive behavior.
These are my suggestions for the authors regarding the manuscript:
- The authors should remove the Material & Methods section line 109-122
- The title is quiet long, I would recommend shortening the title.
- The authors investigated an important topic as many countries are facing vaccine hesitancy. I would recommend that the authors include a few sentences concerning the vaccination strategy in Taiwain. The authors should also discuss if cultural aspects could influence the outcome of their study.
Author Response
Comment 1
The authors should remove the Material & Methods section line 109-122.
Response
Thank you for your reminding. We deleted these sentences “Material & Methods section…” from the revised manuscript. Please refer to line 119. It seems the error occurred in transforming template by the Journal.
Comment 2
The title is quiet long, I would recommend shortening the title.
Response
Thank you for your suggestion. We replaced the original title “Differences in the Constructs of the Extended Protection Motivation Theory Between People With Various Latent Classes of Motivation to Get Vaccinated and to Adopt Preventive Behaviors Against COVID-19 in Taiwan: A Latent Profile Analysis” (34 words) into the new one “Differences in the Protection Motivation Theory Constructs Between People With Various Latent Classes of Motivation for Vaccination and Preventive Behaviors Against COVID-19 in Taiwan” (24 words). Please refer to line 2-5.
Comment 3
I would recommend that the authors include a few sentences concerning the vaccination strategy in Taiwan.
Response
Thank you for your suggestion. We added a paragraph describing the current COVID-19 vaccination strategy in Taiwan as below. Please refer to line 288-301.
“The government in Taiwan announced the latest COVID-19 vaccine distribution strategy on June 21, 2021 by distributing the limited COVID-19 vaccines to the 10 priority groups, with first 3 groups to receive doses immediately; the 10 approved priority groups are: healthcare workers (Group 1), central and local government epidemic prevention personnel (Group 2), frontline workers with a high risk coming into contact with COVID-19 (Group 3), those who need to travel abroad (Group 4), law enforcement officers and firefighters (Group 5), volunteers, long-term caretakers, and care recipients at social welfare organizations (Group 6), national security personnel (Group 7), adults ages 65 and above (Group 8), adults ages 19 to 65 with life-threatening conditions, rare diseases, or a history of serious illness (Group 9), and adults between the ages of 50 and 64 (Group 10) [64]. It indicates that many people may not obtain the protection of vaccination against COVID-19 until their priority to get vaccination. It is important for all people to continue implementing the recommended protective behaviors to reduce the risk of contracting COVID-19.”
Comment 4
The authors should also discuss if cultural aspects could influence the outcome of their study.
Response
Thank you for your suggestion. In Discussion section of the revised manuscript, we added discussion about the experiences of mass vaccination programs against seasonal influenza and hepatitis B in Taiwan as below. Please refer to line 263-266.
“Taiwan has succeeded in mass vaccination programs against seasonal influenza [61] and hepatitis B [62]. Lessons learned from the previous mass vaccination planning may provide recommendations for enhancing preparedness for the COVID-19 emergency that requires mass vaccination.”
Reviewer 2 Report
I have read carefully the content of the article entitled "Differences in the Constructs of the Extended Protection Motivation Theory Between People With Various Latent Classes of Motivation to Get Vaccinated and to Adopt Preventive Behaviors Against COVID-19 in Taiwan: A Latent Profile Analysis". The article raises very important issues of COVID-19 protection among the people of Taiwan. The documet is very interesting and can be an important voice in the discussion on people's attitudes towards the SARS-Cov-2 pandemic. The article presents the discussed issues in a synthetic, concise and understandable way. The extensive bibliography deserves special recognition. However, authors should check the content of the article very carefully, especially lines 109-122. Additionally, the abstract does not have the typical content layout for this type of publication. It is worth the authors to consider redesigning it. Moreover, the summary section is in my opinion far too short compared to the rest of the article. Considering the fact that in the discussion section, the authors raise some interesting threads, it would also be worth considering referring to them in the summary section.
Author Response
Comment 1
Authors should check the content of the article very carefully, especially lines 109-122.
Response
Thank you for your reminding. We deleted these sentences “Material & Methods section…” from the revised manuscript. Please refer to line 119. It seems the error occurred in transforming template by the Journal.
Comment 2
The abstract does not have the typical content layout for this type of publication. It is worth the authors to consider redesigning it.
Response
Thank you for your comment. This journal requires the Abstract presented in an unstructured pattern; therefore, we did not use the format of “Background (or Aims), Methods, Results, and Conclusion.” However, we revised our abstract as below to make it easier to read. Please refer to line 29-38.
“…The present study identified three latent classes, including the participants with high motivation for vaccination and preventive behaviors (the class of Both High), those with low motivation for vaccination and preventive behaviors (the class of Both Low), and those with high motivation for vaccination but low for preventive behaviors (the class of High Vaccination but Low Preventive Behaviors). Compared with the participants in the class of Both High, participants in the class of Both Low had lower levels of perceived vulnerability, perceived severity, self-efficacy to have vaccination, response efficacy of vaccination, knowledge about the vaccination, and previous vaccination for seasonal influenza; participants in the class of High Vaccination but Low Preventive Behaviors had lower levels of perceived vulnerability and perceived severity but higher levels of response cost of vaccination. We concluded that…”
Comment 3
Moreover, the summary section is in my opinion far too short compared to the rest of the article. Considering the fact that in the discussion section, the authors raise some interesting threads, it would also be worth considering referring to them in the summary section.
Response
Thank you for your suggestion. We added the content of Conclusion section as below. Please refer to line 353-360.
“Given that lack of vaccination and protective behaviors may put the people with low motivation for getting vaccinated and adopting preventive behaviors at the highest risk of contracting COVID-19, intervention programs should focus on enhancing their threat appraisal toward COVID-19 and coping appraisal toward vaccination. For people with high motivation to get vaccinated but low motivation to adopt preventive behaviors against COVID-19, enhancing their threat appraisal toward COVID-19 and discussing the advantages and disadvantages of vaccination against COVID-19 are necessary to ensure their vaccination against COVID-19.”
Reviewer 3 Report
- Why Protection Motivation Theory was chosen as a framework while Health Belief Model is more appropriate for this study? The authors should explain the rationale for usage of the chosen model/theory.
- The information on quality of the research tools is lacking in the Method section. The authors used previously constructed tool, they should give the reader information on it`s quality without need to find other article to check the quality of the tool.
- Did the authors assessed the quality of the tools in this study?
- Which knowledge the participants got about vaccination? Type vaccine? Effectiveness? Length of the immunity? Otherwise it looks like participants were asked about some hypothetical vaccine without any information about it, while this information is crucial, according to the model, כor creating איק attitudes towards vaccination. The issue of knowledge given to the participants should be explained in the Methods.
- Limitation part should include the limitation of the choden theory, as this theory does not consider all of the environmental and cognitive variables and the impact of social norms.
- Conclusions are insufficient: authors should explain, which benefits for health promotion may be achieved following the results of this study.
- In the Discussion section authors should explain, in which extent motivation is related to behavior, based on previous studies. What does it mean - high motivation? - based on other studies, does high motivation predict behaviour?
Author Response
Comment 1
Why Protection Motivation Theory was chosen as a framework while Health Belief Model is more appropriate for this study? The authors should explain the rationale for usage of the chosen model/theory.
Response
Thank you for your comment. Both Protection Motivation Theory (PMT) and Health Belief Model are the theoretical frameworks that have been commonly applied to explain the motivation and behaviors regarding vaccination for respiratory infectious diseases. We explained why we chose PMT as the theoretical framework as below. Please refer to line 81-91.
“Protection Motivation Theory (PMT) is one of the theoretical frameworks that have been commonly applied to explain the motivation and behaviors regarding vaccination for respiratory infectious diseases [32-38]. PMT consists of two cognitive processes: threat appraisal and coping appraisal [35]; both have essential roles in determining the level of motivation to adopt certain behaviors to protect individuals from contracting respiratory infectious diseases [36]. Threat appraisal is composed of the perceived severity of and vulnerability to respiratory infectious diseases; coping appraisal is composed of perceived self-efficacy, response efficacy, and cost efficacy of adopting protective behaviors [36,37,39]. The result of a review study supported the effectiveness of Protection Motivation Theory, Health Belief Model, and Theory of Planned Behavior on informing infectious disease modelling, research and public health practice [40].”
Comment 2
The information on quality of the research tools is lacking in the Method section. The authors used previously constructed tool, they should give the reader information on it`s quality without need to find other article to check the quality of the tool.
Response
Thank you for your reminding. We revised the content of the paragraph introducing the research tool by adding the information about the psychometrics of the tools as below. Please refer to line 141-151.
“We measured the constructs of threat appraisal (perceived severity of COVID-19 and perceived vulnerability to COVID-19) by the questionnaire transformed from the questionnaire for measuring risk perception of the H1N1 influenza [53]. A previous study using this questionnaire found that individuals who adopted protective behaviors against COVID-19 had higher levels of threat appraisal than who did not (p < 0.001) [54] and supported its discriminant validity. We measured the constructs of coping appraisal (self-efficacy of receiving COVID-19 vaccination, response efficacy of COVID-19 vaccination, response cost of COVID-19 vaccination, and knowledge about COVID-19 vaccination) by the items of the Drivers of COVID-19 Vaccination Acceptance Scale (DrVac-COVID19S) [55]. A previous study found that the DrVac-COVID19S had acceptable validity and could quickly capture the individuals’ intrinsic intention get COVID-19 vaccination [55].”
Comment 3
Did the authors assess the quality of the tools in this study?
Response
Thank you for your reminding. We added the values of internal reliability of the tool in this study as below. Please refer to line 151-153.
“The Cronbach α of threat appraisal and coping appraisal of this study was 0.704 and 0.821, respectively.”
Comment 4
Which knowledge the participants got about vaccination? Type vaccine? Effectiveness? Length of the immunity? Otherwise it looks like participants were asked about some hypothetical vaccine without any information about it, while this information is crucial, according to the model, כor creating איק attitudes towards vaccination. The issue of knowledge given to the participants should be explained in the Methods.
Response
Thank you for your comment. Firstly, we reviewed the original instrument (DrVac-COVID19S) and found that the items assessed “knowledge about COVID-19 vaccination” but not “knowledge about the mechanism.” We corrected it in the revised manuscript. Please refer to Table 1. Second, we agreed that the knowledge about the effectiveness and complications of various vaccines against COVID-19 may influence the individuals’ motivation to get vaccinated. However, we did not survey them in the present study. We added it as one of the limitations of the present study as below. Please refer to line 340-345.
“Fourth, we assessed participants’ knowledge about COVID-19 vaccination using the items on the DrVac-COVID19S; however, participants’ accuracy of knowledge about the mechanism of COVID-19 vaccination could not be confirmed. Further study warrants to examine the individuals’ knowledge about the effectiveness and complications of various vaccines against COVID-19 and their associations with the motivation to get vaccinated.”
Comment 5
Limitation part should include the limitation of the chosen theory, as this theory does not consider all of the environmental and cognitive variables and the impact of social norms.
Response
Thank you for your reminding. We added it as one of the limitations of the present study as below. Please refer to line 333-340.
“Third, we chose PMT as the theoretical framework to understand the participants’ appraisal and coping appraisal of vaccination against COVID-19; however, we did not evaluate the impacts of environmental variables such as social norms and government policies. Moreover, follow-up studies are needed to examine the psychological, socioeconomic and political moderators and mediators of the association between motivation to and actual behaviors of vaccination against COVID-19. For example, research demonstrated that socio-economic factors influenced the public's willingness-to-pay for the vaccine [71].”
Comment 6
Conclusions are insufficient: authors should explain, which benefits for health promotion may be achieved following the results of this study.
Response
Thank you for your suggestion. We added the content of Conclusion section as below. Please refer to line 353-360.
“Given that lack of vaccination and protective behaviors may put the people with low motivation for getting vaccinated and adopting preventive behaviors at the highest risk of contracting COVID-19, intervention programs should focus on enhancing their threat appraisal toward COVID-19 and coping appraisal toward vaccination. For people with high motivation to get vaccinated but low motivation to adopt preventive behaviors against COVID-19, enhancing their threat appraisal toward COVID-19 and discussing the advantages and disadvantages of vaccination against COVID-19 are necessary to ensure their vaccination against COVID-19.”
Comment 7
In the Discussion section authors should explain, in which extent motivation is related to behavior, based on previous studies. What does it mean - high motivation? - based on other studies, does high motivation predict behaviour?
Response
Thank you for your suggestion. Although no cutoff for the level of motivation predicting further vaccination behaviors, the results of the follow-up studies have confirmed the prediction of motivation to get vaccinated for further actual behaviors of vaccination. We added discussion as below into the revised manuscript. Please refer to line 238-245.
“Research has supported the prediction of motivation to get vaccinated for further actual behaviors of vaccination. For example, a one-year follow-up study confirmed that parents’ lower intention to have their children vaccinated significantly predicted the less likelihood of influenza vaccination among children [59]. A nine-month follow-up study confirmed that higher vaccination intention predicted greater vaccination uptake in healthcare workers [60]. Therefore, enhancing people’s motivation to get vaccinated and to adopt preventive behaviors against COVID-19 is important to stop the spreading of COVID-19.”
Reviewer 4 Report
I am very sympathetic to the paper. It if, needless to say, timely and very important. On this point, I believe, however, that a bit more should be said about why it is crucial to better understand vaccine hesitancy --- that is, it will be *the* single most effective preventive measure to combat the disease.
On the case and the date, I believe that they are fine, but more context should be provided. On the case study, more information about the public health guidance or law should be included. On the data, we should have a better sense of the characteristics of the sample. It is ok to refer to a published study, but still, minimal amount of information should be included.
On the measures: I have one major concern, but luckily, I believe that it will be easy to address. The elephant in the room is the social desirability bias. It should be discussed. Among others, I believe that the authors can acknowledge that it skews their distributions, that is, people are biased. However, this bias should not affect the inferential purpose of the authors. Why? Because we know that the bias is homogenous across subgroups, according to studies by Daoust et al. (2020, in Journal of Experimental Poli Sci) and Daoust et al. (2021 in PLOS One).
Section 2: the authors did not remove the instructions for Material and Methods (lines 109-121).
The findings are quite straightforward, elegant, and improve our understanding of the issue. However, I think more should be done when discussing the findings for age. Why? It is a very important variable, especially linked to COVID-19 hospitalization and deaths, which makes it particularly important to understand. See, among others, Daoust (2020 in PLOS One).
Author Response
Comment 1
A bit more should be said about why it is crucial to better understand vaccine hesitancy --- that is, it will be *the* single most effective preventive measure to combat the disease.
Response
Thank you for your suggestion. We added the importance of vaccination for stopping the spreading of COVID-19 as below into the revised manuscript. Please refer to line 48-49.
“An efficacious vaccine is considered essential to prevent further morbidity and mortality of COVID-19 infection [9].”
Comment 2
On the case study, more information about the public health guidance or law should be included.
Response
Thank you for your suggestion. Although no vaccine against COVID-19 was available in Taiwan during the study period (October 15 to December 21, 2020), the government in Taiwan proposed the current COVID-19 vaccination strategy under continuous revision as the COVID-19 pandemic and the doses of available vaccines change. We added a paragraph describing the current COVID-19 vaccination strategy in Taiwan as below. Please refer to line 288-301.
“The government in Taiwan announced the latest COVID-19 vaccine distribution strategy on June 21, 2021 by distributing the limited COVID-19 vaccines to the 10 priority groups, with first 3 groups to receive doses immediately; the 10 approved priority groups are: healthcare workers (Group 1), central and local government epidemic prevention personnel (Group 2), frontline workers with a high risk coming into contact with COVID-19 (Group 3), those who need to travel abroad (Group 4), law enforcement officers and firefighters (Group 5), volunteers, long-term caretakers, and care recipients at social welfare organizations (Group 6), national security personnel (Group 7), adults ages 65 and above (Group 8), adults ages 19 to 65 with life-threatening conditions, rare diseases, or a history of serious illness (Group 9), and adults between the ages of 50 and 64 (Group 10) [64]. It indicates that many people may not obtain the protection of vaccination against COVID-19 until their priority to get vaccination. It is important for all people to continue implementing the recommended protective behaviors to reduce the risk of contracting COVID-19.”
Comment 3
On the data, we should have a better sense of the characteristics of the sample. It is ok to refer to a published study, but still, minimal amount of information should be included.
Response
Thank you for your reminding. We agreed that personal characteristics of the participants, for example, socioeconomic status should be investigated to examine their roles for motivation to get vaccinated. We added it as one of the limitations of the present study as below. Please refer to line 333-340.
“Third, we chose PMT as the theoretical framework to understand the participants’ appraisal and coping appraisal of vaccination against COVID-19; however, we did not evaluate the impacts of environmental variables such as social norms and government policies. Moreover, follow-up studies are needed to examine the psychological, socioeconomic and political moderators and mediators of the association between motivation to and actual behaviors of vaccination against COVID-19. For example, research demonstrated that socio-economic factors influenced the public's willingness-to-pay for the vaccine [71].”
Reviewer 5 Report
The article is current as we can consider that we are still at risk of an epidemic and that we can become infected and spread the disease, in addition to the fact that we do not have a population sufficiently vaccinated to reduce the spread of the disease in the world, requiring non-pharmacological protective measures
The study makes clear the three different classes of people motivated to get vaccinated, comply with protective measures or not. However, it cannot make a strict correlation of those who have no motivation. It will be necessary to deepen the research by extending the methodology with some more socioeconomic, sociological and political variables to clearly define the condition that most interferes with motivation. Recruitment may have been a point in the methodology that compromised the result, as those who, for some reason or curiosity, wanted to respond via Facebook. Even so, I consider an article of current interest and it is possible to extrapolate to other universes that also suffer from the denial and the behavior of its own citizen
Author Response
Comment 1
The study makes clear the three different classes of people motivated to get vaccinated, comply with protective measures or not. However, it cannot make a strict correlation of those who have no motivation. It will be necessary to deepen the research by extending the methodology with some more socioeconomic, sociological and political variables to clearly define the condition that most interferes with motivation.
Response
Thank you for your reminding. We agreed that further studies, especially follow-up studies are needed to examine the psychological, socioeconomic, and political moderators and mediators of the association between motivation to and actual behaviors of vaccination against COVID-19. We added it as one of the limitations of the present study as below. Please refer to line 333-340.
“Third, we chose PMT as the theoretical framework to understand the participants’ appraisal and coping appraisal of vaccination against COVID-19; however, we did not evaluate the impacts of environmental variables such as social norms and government policies. Moreover, follow-up studies are needed to examine the psychological, socioeconomic and political moderators and mediators of the association between motivation to and actual behaviors of vaccination against COVID-19. For example, research demonstrated that socio-economic factors influenced the public's willingness-to-pay for the vaccine [71].”
Comment 2
Recruitment may have been a point in the methodology that compromised the result, as those who, for some reason or curiosity, wanted to respond via Facebook. Even so, I consider an article of current interest and it is possible to extrapolate to other universes that also suffer from the denial and the behavior of its own citizen
Response
Thank you for your comment. Recruiting participants through Facebook inevitably resulted in the problem of sample bias. We have listed it as the major study limitation of this study. Please refer to line 319-324.
“First, this study used a Facebook advertisement to recruit participants. Although delivering study questionnaires through social media can provide a large number of responses in a short period [67], particularly during the pandemic, the problem of sample bias is inevitable [68]. For example, Facebook users consist of younger and more progressive people among the general population [58].”